# Qualitative Study of Maternity Healthcare Vulnerability Based on Women’s Experiences in Different Sociocultural Context

**DOI:** 10.3390/nursrep15030105

**Published:** 2025-03-18

**Authors:** Claudia Susana Silva-Fernández, Eva Garrosa, David Ramiro-Cortijo

**Affiliations:** 1Department of Biological & Health Psychology, Faculty of Psychology, Universidad Autónoma de Madrid, 28049 Madrid, Spain; 2Instituto Universitario de Estudios de la Mujer (IUEM), Universidad Autónoma de Madrid, 28049 Madrid, Spain; 3Department of Physiology, Faculty of Medicine, Universidad Autónoma de Madrid, 28029 Madrid, Spain

**Keywords:** qualitative methods, maternity healthcare, vulnerability, health institutions, health professionals, women empowerment

## Abstract

**Background:** Unfulfillment of maternity rights in healthcare is a global problem associated with abuse, neglect and discrimination, known as obstetrics and gynecology (OB/GYN) vulnerability. Women’s perceptions of their experience are a keystone to improving maternity healthcare. The aim of this study was to evaluate the women’s perceptions of the vulnerability of maternity rights and the associated risk and protective factors. **Methods:** This study was carried out by qualitative techniques based on the analysis of a semi-structured interview applied to six women in the postpartum period with pregnancy assistance and birth in Spain and Colombia between February and August of 2024. A triangulation analysis was performed about the perceptions of the concept, experiences and risk and protective factors of OB/GYN vulnerability. The free-access ATLAS.ti software was used. **Results:** OB/GYN vulnerability is generally perceived by women with a psychological impact. Women think that their own factors (emotion management, social support, attitude to change and beliefs), health professional factors (burnout, empathy and social skills) and health institution factors (workload, centralization in technical and protocols, humanization, quality and access to recourses) have an influence to modulate the vulnerability of rights in maternity healthcare. **Conclusions:** It is necessary for health systems to move from a protocol-centered to a person-centered model, particularly in maternity healthcare. This model should include the biopsychosocial needs of women and allow for their participation. Health institutions need to evaluate their processes and minimize burnout in health professionals. In addition, there are factors affecting OB/GYN vulnerability not only in childbirth but also during pregnancy and postpartum.

## 1. Introduction

Obstetric and gynecological (OB/GYN) vulnerability is the actions or attitudes of health providers that fail to uphold women’s rights in medical care [1]. OB/GYN vulnerability is represented in situations of abuse (physical, verbal and sexual), negligence, discrimination, inadequate application of techniques and resources [2,3,4]. Autonomy, dignity, privacy, respectful care and information are violated rights by health providers under OB/GYN vulnerability [5,6,7,8]. Colombia describes a 69.0% OB/GYN vulnerability [1], while Spain estimates a 67.4% OB/GYN vulnerability [9].

Psychosocial risk factors predispose OB/GYN vulnerability, such as immigrant, low economic resources and social support, HIV, young age, unemployment and the use of psychoactive substances [4,6,10]. However, protective factors have been identified, such as resilience, positive affect, multiparous, vaginal birth, age over 30 years, awareness and women empowerment [6,11,12]. Another modulatory factor is culture, which promotes the authoritarian, asymmetric relationship between a health provider and a woman, pathologization of childbirth [13], lack of time of healthcare [14], burnout in healthcare providers [15,16,17] and non-individualization of healthcare assistance [1,4].

The World Health Organization (WHO) put forward prevention and research into vulnerable situations of women during childbirth [18]. Furthermore, it has developed not only statements of healthcare maternity rights but also guidelines for a positive experience during pregnancy and childbirth [19]. The main message was that women perceive their childbirth experience as positive when their sociocultural, physical and psychological well-being is preserved [20,21]. In addition, preventive strategies are implemented in health institutions but require evaluation. A key plan would be person-centered care, recognizing the psychosocial needs of women and carrying out an honest communication [7]. Thus, knowing the perception of OB/GYN vulnerability is necessary to explore the experience of women [22]. This background led us to ask about the components of healthcare that should be improved to minimize vulnerability of rights during healthcare in pregnancy, childbirth and postpartum, considering the sociocultural context as a modulating factor of vulnerability [23].

## 2. Materials and Methods

### 2.1. Study Design

The main purpose of this qualitative study was to evaluate the experience and perception of women who gave birth in the last five years. For this purpose, a virtual, in-depth, semi-structured interview was carried out in the two different social contexts of pregnancy assistance and birth (Spain and Colombia). Both sociocultural contexts were chosen considering their health system differences [23]. Interviews and data collection were carried out between February and August of 2024. This study was approved by the Research Ethics Committees of Universidad Autónoma de Madrid (Madrid, Spain; CEI-112-2199, 22 January 2021) and FOSCAL Hospital from Colombia (Santander, Colombia; FOSCAL-06939/2022, 23 September 2022).

This study explored the healthcare experience during pregnancy, childbirth and postpartum and the associated factors to increase OB/GYN vulnerability. To understand these events, it was suggested to study the experience of women [24]. In the perception of events, experience assigns meaning to reality, merging thoughts, feelings and knowledge.

### 2.2. Characteristics of the Women Participants

The six women interviewed had children between 2 months and 4 years old, and they had their last birth in Colombia (C; n = 3) or Spain (S; n = 3). The women were selected through systematic and probability sampling from a cohort of 185 Hispanic-speaking women from a previous study [23], considering moderate scores in the perception of maternity rights but also the availability to carry out the interview. All the women were willing to participate, and the informed consent form was signed in each case.

To characterize the participants, an ad hoc sociodemographic questionnaire was applied to record age, marital status, occupation, context of birth (Colombia or Spain) and type of healthcare center (public or private). Additionally, the women were asked for obstetrical data, such as type of pregnancy (single or multiple), type of delivery (vaginal or C-section) and parity. The age range was 22 to 38 years old (mean = 31; standard deviation = 5.7). Most of the women received care in a public health facility for their delivery between 2019 and 2024. Their characteristics are described in Table 1.

### 2.3. Interview Design

The main instrument was an ad hoc semi-structured interview designed specifically to explore the aim of this article (Appendix A), following the previously published factors associated with obstetric violence [10]. The interview had 12 open questions oriented to two categories of analysis: (1) perception of vulnerability of maternity rights into healthcare during pregnancy, delivery and postpartum; and (2) perception of factors related to OB/GYN vulnerability.

All interviews were audio-recorded, and three of them gave the consent to be video-recorded. The recordings lasted between 26 and 45 min. For privacy and ethical reasons, no video recordings are attached. However, samples of the audio recordings can be downloaded in the Appendix A. According to the ethical policy of the Spanish government, these audio recordings were edited to de-identify personal data. All interviews were supervised by the research team and were conducted by a health psychologist, C.S.S-F. To ensure balanced interpretations, the recording was checked to validate preliminary themes, and peer debriefing sessions with all research teams were held to challenge potential biases.

### 2.4. Data Analysis

To ensure accuracy and detail, all transcription was literally and manually written by the interviewer. Then, the interviews were analyzed by ATLAS.ti (version 22, Scientific Software Development GmbH. Technische Universität Berlin, Berlin, Germany; www.atlasti.com). ATLAS.ti is computer-assisted software that facilitates the analysis of qualitative data for research. In ATLAS.ti, the emerging themes (categories) and subthemes (ideas) were established according to the transcriptions of the women. In addition, to identify the ideas of greatest relevance, we studied the rooting (R; frequency that the concept was related in the discourse of the participants, meaning the higher the R, the more relevant the concept) and the density (D; number of other concepts linked to the analyzed one, meaning the lower the D, the more individualized the concept). It is necessary to note that subthemes can be implemented in more than one theme. The data saturation based on the repetition of the data collected as well as sample equity by birth context were used [25]. Subsequently, to increase the validity and depth of the results, triangulation was used. Triangulation is a strategy to enhance the credibility and reliability of qualitative research. In the present manuscript, data triangulation was involved, gathering data from the six women and two contexts (Colombia and Spain) to ensure understanding. Thus, the interviews with the women and the risk/protective factors of OB/GYN vulnerability were used. The theoretical framework was based on a biopsychosocial model considering risk (medical conditions, anxiety, access to healthcare, etc.) and protective factors (social support, resilience, optimism, etc.). For publication purposes, some participants’ expressions were edited in the results, eliminating repetitions, filler words and false starts of sentences. Additionally, the original Spanish transcript can be consulted in the Appendix A.

## 3. Results

The analysis was based on two key themes. Firstly, the perception of vulnerability is due to the non-fulfilment of maternity rights during healthcare assistance. This involved not only the experiences of vulnerability but also the knowledge related to the rights. The second was the perception of risk and protection factors for OB/GYN vulnerability. The examination was based on a content analysis referring to categories and rooting and density values.

### 3.1. Experience of Vulnerability and Knowledge Related to Maternity Rights

The women determined that OB/GYN vulnerability was a real event that women experience in motherhood. They primarily associated it with psychological impact actions (Rooting (R) = 6, Density (D) = 1), attributing this to a lack of empathy and disrespectful communication regarding women’s needs and conditions. They also associated it with physical violence (R = 2, D = 1) and negligence (R = 1, D = 1).


*“I have heard from women who have many children that physicians begin to generate the rejection that they get pregnant all the time, and they begin to make comments that belittle that condition”.*
(Transcript I—C3).

Physical violence: *“When they [physicians] have to perform labor instrumentation in the ways they do it”*.(Transcript I—S3).

The women perceived two origins of vulnerability, (1) institutional and (2) professional, both associated with the care processes during pregnancy, childbirth and postpartum. Among the vulnerability originated by the institution, the women perceived more in childbirth (R = 11, D = 5) compared to pregnancy (R = 9, D = 5) and postpartum (R = 3, D = 4), while the experiences of vulnerability originated by the actions of the professional were perceived more in pregnancy (R = 22, D = 7) than childbirth (R = 11, D = 6) and postpartum (R = 6, D = 5). The women associated the vulnerability of their rights during healthcare with experiences based on non-compliance with safety, problems in accessing services and professionals as well as the absence of pain relief and performance of uncomfortable techniques. All of these are summarized in the category of failure in the quality and timeliness of care (Figure 1).

Problems with access to healthcare: *“I know many cases where it was 3 or 4 in the afternoon, they had been starving all day, and they had not been taken to an operating room because there was no order”*.(Transcript II—C2).

Problems with professionals: *“The physician who assist me, I don’t know what was on his mind, and he did an ultrasound and told me: no, the baby is growing in the whole scar from the previous C-section, it can’t be born; I think that this week you will have a spontaneous abortion or come a day and I will perform the abortion because it can’t be born. Another doctor saw me, did an ultrasound and told me: no, that there was a 4 mm detachment, but with hormones and care it would be fine. And so, it was”*.(Transcript III—C1).

No compliance with safety: *“I thought, they will induce labor, they will be there observing me, and everything will be very controlled. But it is not like that. My blood pressure dropped, and nobody noticed”*.(Transcript II—S1).

The women perceived negligence in the dignified treatment and autonomy rights by the institution and health professional. This showed the perception of a dynamic interaction between the health provider and woman based on the minimization of capacities and the equal application of women’s rights.

*“They* [physicians/nurses] *talk to you as: it’s your fault, you looked for it. Literally, one day a nurse told me: well, why did you open your legs?* [refers to her pregnancy]*.”… “They* [physicians/nurses] *don’t listen, they don’t pay attention to me simply because: he’s the physician. I listen to my body and understand many things, I know that something is changing and that something is suddenly not right”.*(Transcript III—C2).

*“They said* [physicians/nurses]*: you are fatter than a pig, you can’t be that fat, like a seal. Ugly things!”.*(Transcript III—S1).


*“The right to care anywhere, because some time, I go and need to get a health card-insurance, but I can’t if I don’t have a valid passport, which means a lot of documentation”.*
(Transcript III—S2).

To prioritize interventions, technics and protocols was the most common subtheme belonging to the quality, opportunity of care and autonomy themes. The situations were described as the Kristeller’s procedure, frequent vaginal examinations, instrumentalized birth and the application of protocols, not considering the autonomy of women.

*“Those vaginal examinations that they* [physician] *do to me all the time… the truth was very traumatic… I don’t know if that is medically necessary or if there is another way… that is what I felt like they were putting their hand, fist and elbow inside of me”.*(Transcript IV—C3).

*“As they* [physician/nurse] *are in favor of vaginal birth, they make me endure until I almost die for natural birth occur. But I don’t know, let the mother’s priority prevail”.*(Transcript IV—S1).


*“I pushed, and because of the epidural, I couldn’t apply enough force… The physician had to get on top of me and use their arms to press down on my stomach so that the baby would come out. I remember that as being really awful… if it wasn’t for the baby dying, nobody should do this, because I had a bad time and was uncomfortable feeling and I’m in a beautiful moment and I’m there dying, because it hurts when they press down on your stomach, it doesn’t hurt when you give birth”.*
(Transcript IV—S3).

Lack of empathy by professionals was another subtheme that stood out in the perception of vulnerability described by the women. This belongs to the themes of failures in dignified and respectful treatment, comprehensive and individualized care, quality and timeliness of care, and autonomy. This showed actions and attitudes of the professional in which there is an unfulfillment of rights due to the lack of recognition and understanding of psychosocial elements of women.


*“A nurse said: you abandoned your daughter here. Financially, she didn’t know how I was doing; emotionally, also, she didn’t know how I was. If it would be for me, I would have spent 24 h there. But I had another newborn at home, I didn’t have anyone to take care of her, and I knew that the baby who was in the clinic was surrounded by professionals. I went twice a day. The truth is I felt really bad, I didn’t abandon my daughter, I just couldn’t do it”.*
(Transcript V—C2).

*“They* [nurse] *are like-you have the child, and you have to know-, but you are not born knowing. They should put themselves in the place of the woman who has just given birth or who does not even know how to hold the child”.*(Transcript V—S2).

*“I am extremely worried about my baby because in the hypothetical case that he has Down syndrome, what can I do? Do I continue or not? And the other person* [health professional] *only thinks about the 600 euros for the test”.*(Transcript V—S3).

There were other issues that were identified in the analysis. At the institutional level, the women acknowledged that staff turnover creates a feeling of violation of their privacy due to the frequent exposure of their intimacy. At the professional level, women perceived that there were problems with communication, a lack of attention focused on their needs and even situations of gender harassment.

Gender harassment: *“They* [health professionals] *exceed with women, for example by touching physically or commenting on their physical appearance”*. (Transcript VI—C1).

Lack of communication: *“They* [health professional] *say ‘Get ready to not sleep’* [ironically]*. I say: ‘What?’. I can’t imagine the magnitude of what is coming. They should sit people down and tell them: You are not going to sleep because of this, organize your time, prepare yourself like this, create a routine like this”*. Bad communication: *“He* [physician] *told me: a normal pregnancy is just a baby; this is already starting badly* [multiple pregnancy]*”*.(Transcript VI—C2).

Bad communication: *“The first time I tested positive for toxoplasmosis, she* [physician] *could not explain it to me, and the truth is I was scared, I cried”*.(Transcript VI—C3).

*“The bad thing is that I did not have a person to trust, to go to the same professional, and I can tell them. They are guided by what the previous one has written”.* Lack of attention: *“I asked: please, I feel terrible, if I have already dilated completely and the baby has not come down, is there any solution? As if they* [health professionals] *hadn’t heard me”*.(Transcript VI—S1).

### 3.2. Perception of Psychosocial Factors Influencing OB/GYN Vulnerability

The interaction systems of OB/GYN vulnerability were analyzed to identify risk and protective factors by institution, health professional and woman. The analysis showed that the healthcare institution and professionals can directly affect the woman, with the institution also exerting indirect influence (Figure 2). The women recognized more risk (R = 69, D = 3) than protective factors (R = 52, D = 3). Furthermore, they highlighted the factors associated with women (Risk: R = 41, D = 14; Protective: R = 21, D = 12).

The analysis of the women’s characteristics showed risk factors associated with difficulty in managing internal (beliefs, knowledge, emotions) and external resources (support, economic) to cope with the changes in motherhood. The women thought that access to unreliable information on the internet, especially social networks; social pressure; socioeconomic problems; lack of knowledge; the physical and psychological impact of motherhood and the fluctuation of life are elements that affected them during healthcare.

Lack of maternity and hospitalization experience and low socioeconomic conditions: *“In girls or very young women who become pregnant or who live in a village, who know very little”*.(Transcript VII—C1).

Expectations and beliefs of idealization of motherhood: *“Motherhood was the most beautiful thing in the world, but no, postpartum depression exists. Motherhood has been romanticized a lot”*. Changes due to motherhood: *“It is very hard, it is very heavy, it is three months in which you do not sleep, literally I did not sleep at all, and I have to take care of two babies, when many times I do not even know how to take care of yourself”*.(Transcript VII—C2).

Expectations and beliefs of motherhood: *“I think that everything is automatic when you have your first child, but no, there are things that are definitely enough complex”*.(Transcript VII—C3).

Emotional lability and alteration of physical state: *“You have the doors open* [in the healthcare center]*, what am I hearing? -Delivery room number I don’t know what, running to the operating room- it makes me even more anxious because I don’t feel well, I was afraid because I didn’t know what was really going to happen, and I was hearing external things, and I didn’t know if that was going to happen to me too”*. Inconsistent information: *“When I was pregnant, it seems like everyone tells me something different, I really don’t know what to believe”*.(Transcript VII—S1).

Lifestyle changes by motherhood: *“I put aside my own things, the professional side in my case, and to ensure that they* [children] *are not left alone or are not in daycare or with other people all day, it is a bit of a sacrifice”*.(Transcript VII—S2).

Emotional lability: *“I have my emotions so on edge, it’s uncontrollable”*.(Transcript VII—S3).

The women perceived a positive relationship with the health professional, a positive attitude towards changes in motherhood, internalization of interests, social support, having private health insurance and knowledge of previous experiences as protective factors of OB/GYN vulnerability. It is important to note that positive attitude was associated with emotional coping, willingness to change and the implementation of distracting behavioral patterns.

Positive attitude: *“I didn’t have a bad life. Many factors depended on my emotional state, on whether my daughters were well. I tried to be as calm as possible; I helped the nurses; I helped them do things because I had nothing else to do* [during hospitalization]*”*.(Transcript VIII—C2).

*“I had the premium plan* [health insurance]*, I had direct access to the gynecologists and obstetricians. I didn’t have to go around in circles as I know other women sometimes have to”.* Prioritization to women: *“That I can have a phone number, or a pediatrician, or a specific gynecologist-obstetrician… something much closer than schematized, than having to make an appointment and having to travel. Sometimes so much procedure is a bit complex for the situation of a state of pregnancy”*. Information, education and assertive communication: *“Breastfeeding was not easy either… the advice from the nurses because at first I did not have any milk, that really helped me a lot… they teach you how to do it”*. Family support: *“They* [health institution] *allowed my husband to be present at every moment of the process, including allowing him to enter the labor room without problems”*.(Transcript VIII—C3).

Motivation: *“It was a process that is an insemination* [her pregnant]*, and I achieve it, of course it is the greatest happiness. And then if I have a super good pregnancy, which I didn’t vomit, I could live a normal life. So, I am much more motivated because I can live my normal life and added to that I was going to have a baby”*.(Transcript VIII—S1).

Active participation during care: *“You have to talk and say what you feel as a mother and be taken into account”*. Adherence to the recommendations of health professionals: *“The willingness to change my routine a little. When I was pregnant, I have to do a number of things, both with regard to food and taking medicine”*.(Transcript VIII—S2).

Previous experience: *“I was not very nervous, and I was more aware of everything that was happening”*. Social support: *“A friend who works in a hospital told me: come and we’ll talk to the gynecologist and see what happens”*. Constant monitoring: *“They* [health institution] *sent me to a physician. Then, I had a couple of problems with the baby because she had epileptic seizures, and the physician immediately sent me to a psychologist, and they have been treating me very well”*.(Transcript VIII—S3).

Regarding the women’s perceptions of professional risk factors, the most prevalent were associated with poor social communication skills and empathy as well as difficulties in emotional regulation. The women perceived attitudes that limited compliance with person-centered care protocols.

Poor emotional regulation: *“If you work under stress… in the end you pay with the least guilty person in very delicate situations, because childbirth is a situation that you have to take carefully. Each case is a different story, and you cannot treat people badly because you are having a bad day* [regarding health professionals]*”*. Non-compliance with care protocols: *“They* [nurse] *just suture that up, they didn’t check it* [vagina]*. The gynecologist looked at me before discharging me: this is perfect; -how perfect?—Then I was worried”*.(Transcript IX—S1).

Lack of empathy: *“They* [health professionals] *should have a little more empathy… to put themselves in the place of the woman who has just given birth and teach her”*.(Transcript IX—S2).

Communication skills deficiencies: *“The midwife would come and say: I like quick and painless births… and I would say: me too. He wouldn’t say anything else. And I would only stick with quick and painless”*.(Transcript IX—S3).

The analysis of the protective factors of professionals allowed for the identification of the importance of professional assistance to the needs, recognizing not only their biological component but also their beliefs, values and feelings. Likewise, this requires that the professionals be flexible and carry out assertive communication.

*“Each patient needs to receive more personalized attention because each case is different. Maybe what I need someone else doesn’t need. They* [physician] *can’t treat everyone the same”.*(Transcript X—S1).

Participation of women in its care: *“They [health professionals] said to me: do you want to see it? Me: yes. They put a mirror; I could see it… it was fantastic. In fact, they said to me: when the baby is coming out, put your hands out so you can hold it. As soon as the baby came out, I held her with my hands and put her on my chest. I have the whole super nice memory”*.(Transcript X—S3).

Furthermore, the women perceived that the overload of services, the salary and workday conditions of health professionals are key institutional elements that facilitate the unfulfillment of the women’s rights during healthcare.

Poor working conditions: *“Maybe, they* [physician] *get paid too late… or because the workday is too long, and they are tired. If their rights are violated, they are not going to be treated with the best attitude”*.(Transcript XI—C2).

Lack of humanization: *“They should be trained a little better in care… a module on humanity, on how to treat the person from a psychological, sensitive point of view, beyond the procedure”*.(Transcript XI—C3).


*“I also expected that the same specialists would follow up with you until the end, but that is not true”.*
(Transcript XI—S1).

Lack of organization: *“It is true that they* [health institutions] *put a lot of pressure on the professional: -you have to attend to so many patients a day-… they have to hire more physician, because in the end, they* [physician] *cannot do miracles and they cannot attend to you in 1 min, it is impossible”*.(Transcript XI—S3).

The women perceived that early and constant care was the main factor of the health institution to minimize OB/GYN vulnerability. Other protective variables were assertive communication and inclusion of women and families in care, besides flexibility, comfort, punctuality and quality of the health services.

## 4. Discussion

The main result of this article supports that the vulnerability of maternity rights during healthcare is a real phenomenon experienced by women regardless of the health institution (public and private) and social context. Vulnerable (practices or decisions that may put patients at risk due to a lack of safeguards, oversight or proper protocols) or omission actions (failures to act when necessary) are associated more with psychological impact than with physical or sexual impacts. Previous studies reported over 50% of abuse and violation of maternity rights in health services [1,9,26,27]. Also, the Special Rapporteur on violence against women of United Nations Organizations supports the high global frequency of OB/GYN vulnerability, highlighting the psychological impact [28].

The women perceived OB/GYN vulnerability to be carried out not only by health professionals but also by health institutions. Moreover, this phenomenon was more frequent in pregnancy and childbirth. Consistent with this, there are studies that recognize health institutions as a generator of vulnerability [29], with the main sources of vulnerability being health services that affect privacy and inhibit women’s participation and insufficient professionals or low-quality technical services [1,2,29]. It has been documented that the characteristics and quality of the relationship between a professional and woman can influence access to health services [30], the well-being of a woman [31], the degree of satisfaction with the services [32] and the perception of vulnerability of rights [10].

The present data show that difficulties in the quality and timeliness of care had an impact on the perception of OB/GYN vulnerability. This perception was exposed in situations where there was no comprehensive care, was harmed from the incorrect application of techniques or diagnoses and were limitations in access to services. Other authors had linked excessive medicalization and pathologization [33], late and ineffective care [34] and unjustified and painful invasive practices [1,4,35] as actions that violate women’s rights. These actions have been justified by an industrialized model: standardization of processes, fragmentation of mind and body, prioritization of economic stability of health organization and lack of interest in the subjective relationships [13] have been its justifications.

According to our data, complications to dignified and respectful treatment were associated with a lack of empathy, verbal abuse and discrimination. In turn, the exclusion of women, disregarding their voices and prioritizing institutional protocols over their interests, feelings and beliefs were associated with failures in autonomy. For health professionals, childbirth is a routine process, but for women, it is a significant experience [36]. Thus, healthcare during childbirth must be compassionate, careful, effective and respectful of the woman and promote autonomy [37]. Autonomy is a fundamental right to be fulfilled during maternity healthcare; women can adapt to motherhood according to their expectations and capacities.

Researchers have pointed out that health professionals can lose their sensitivity and empathy due to burnout in the work caused by stressors such as heavy workload, being underpaid, lack of resources and routine processes [4,15,16,17]. In the health service context, professional stress has been proposed as a vicious cycle of arousal-related behaviors for vulnerability of the rights of users [38]. These results were consistent with the behavioral model, identifying, in the institutions, the unfavorable working conditions, and, in the professional, the poor emotional regulation of stress, as being the main risk factors. Thus, the difficulty for healthcare is not only the humanization for women but also the humanization of working conditions in healthcare itself.

These data also support that the experience of women related to ambiguous and deficient information by healthcare professionals contributes to GYN/OBS vulnerability. The changes in motherhood generate the need to develop skills for women, with information being a primary resource. Adequately informing the women about breastfeeding or newborn care has shown a favorable impact on the woman’s adaptation during pregnancy and postpartum [39,40]. Thus, communication based on clear information during healthcare strengthens women’s empowerment and participation [7,8]. The practice of patient-centered care implies that, from the prenatal period, communication must be open, constant, honest and respectful between women and health professionals, controlling the asymmetry in the relationship [7]. Strengthening the communication skills of health professionals is important because these skills allow for them to establish trust and to be effective at transmitting messages and calm [41].

Finally, the present data support that women perceived internal resources to cope with the changes in motherhood as risk or protection factors for OB/GYN vulnerability. These include beliefs about motherhood, willingness to change, motivation, knowledge acquired from previous motherhood and emotional management. In line with this, other researchers pointed out that the women’s attitudes related to maternal experience evolve their beliefs about the mother’s role [42] and their perception of the context and their own resources [43]. It has been documented that, in pregnancy, women feel emotionally fragile due to the imbalance in the perception of control and fulfillment of expectations [36,44]. Additionally, in the first weeks postpartum, women often feel disorientated about caring for a newborn, which can change their emotional state [45], and to be resilient to motherhood adaptation [46,47], preventing problems in mental health [48]. In this study, women also associated external factors, such as social support, and socioeconomic changes that motherhood brings. In relation to this, other studies have found that family and social support contributes to coping [47], preventing mental health disorders during motherhood [10,49,50]. Furthermore, some women experience dissonance between the ideal and real motherhood due to giving up other roles [51]. Thus, for women, empowerment needs to consider that the perception of the motherhood can be influenced by multiple determinants, such as historical, biological, psychological and socioeconomic [47].

### 4.1. Implication for Healthcare Practice

Among the most important things to consider is the re-evaluation of the professionals to avoid the use of painful and unnecessary techniques, which the WHO no longer recommends [20]. In addition, health institutions could consider changes in the management of human resources, applying strategies to minimize stress and overload in professionals. The application of checklists on protocol compliance and rights during maternity healthcare [15] has been implemented as an adequate solution.

Furthermore, it would be necessary to promote a person-centered care approach, assessing biological, psychological and social needs of women. To this end, psychosocial professionals can be integrated into the maternal healthcare team; also, the education of professionals in social skills and primary prevention of psychosocial risk conditions can be strengthened. Other authors have suggested health professionals should receive training in evidence-based clinical practices, social skills and fundamentals in gender and rights [3]. Another key element would be involving the woman in her care process. This includes adequate information, feelings and beliefs as well as decision-making. At the same time, it is emphasized that family and social support provides resources to women with a low experience in motherhood. This recommendation is complementary to the ones already given by experts, including attention to the beliefs and socio-emotional needs of women [7,11].

### 4.2. Strengths and Limitation

This study contributes to exploring other variables that influence OB/GYN vulnerability that are less addressed. Generally, studies have focused on the characteristics of health professionals rather than delving into the institution and the woman. Also, this research analysis is not only focused on pregnancy but also the postpartum period in which women’s rights could be violated during maternity healthcare, since other studies have generally focused on childbirth.

However, as a limitation, this research analyzed the subjectivity of only six women, thus being difficult to generalize the results. In addition, a recall bias is likely possible due to some women being interviewed 3 years after postpartum.

## 5. Conclusions

OB/GYN vulnerability is commonly perceived by women in maternity healthcare, which occurs regardless of social context and type of institution (public or private). Women’s perception is that it prioritizes the comfort of the professional and the economic and technical security of the institution instead of the needs of the women. In addition, women perceive a lack of empathy and respect for the maternity assistance. Women perceive that improving working conditions of professionals, availability of quality resources, promotion of social support, continuous monitoring and non-discriminatory care are key elements within healthcare institutions to minimize OB/GYN vulnerability. Similarly, emotional regulation, communication skills and humanized care by healthcare professionals are crucial. For women, key factors to prevent vulnerability include previous experiences, beliefs, emotional regulation, social support and adherence to recommendations. Health institutions should implement changes in the training of professionals and treatment, recognizing the biopsychosocial components and the importance of respect and participation of women.

## Figures and Tables

**Figure 1 nursrep-15-00105-f001:**
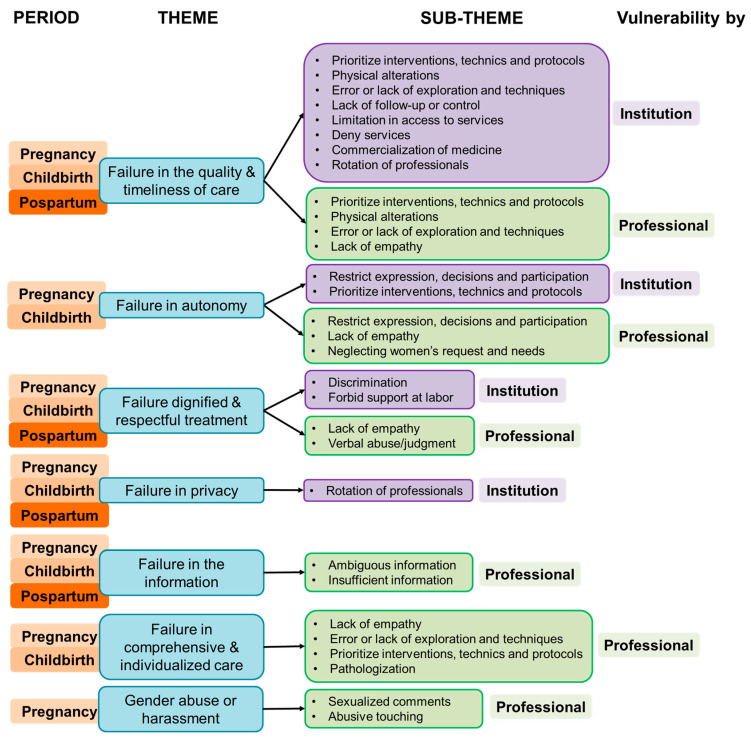
The perception of failures influencing OB/GYN vulnerability differentiated by institution and professional and affects by gestational period.

**Figure 2 nursrep-15-00105-f002:**
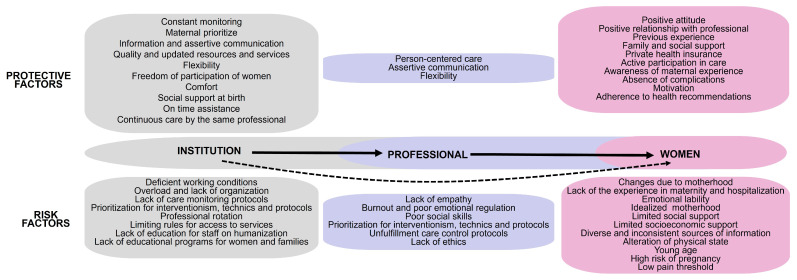
Women’s perceptions of vulnerability risk and protective factors clustered by health institution, professional and women. The solid arrow indicates direct effects, and the dotted arrow indicates indirect effects.

**Table 1 nursrep-15-00105-t001:** Sociodemographic and gynecological characteristics of the women interviewed.

Women	Context	Status	Age	Occupation	Healthcare Center	Pregnancy	Labor	Parity
C1	Colombia	Married	28	Employed	Public and Private	Single	C-section in 2022	2
C2	Colombia	Single	22	Unemployed	Public	Multiple	C-section in 2019	1
C3	Colombia	Married	35	Employed	Public and Private	Single	Vaginal in 2022	1
S1	Spain	Single	38	Employed	Public	Single *	Vaginal in 2019	1
S2	Spain	Married	30	Unemployed	Public	Single	C-section in 2023	2
S3	Spain	Married	33	Employed	Public and Private	Single	Vaginal in 2024	2

Age and occupation were extracted at labor. * Assisted reproduction techniques.

## Data Availability

The data presented in this study are available upon request from the corresponding author. The availability of the data is restricted to investigators based in academic institutions.

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
