# Peer review of "Qualitative Study of Maternity Healthcare Vulnerability Based on Women’s Experiences in Different Sociocultural Context"

_nursrep, 2025, doi:10.3390/nursrep15030105_

Round 1

Reviewer 1 Report

Comments and Suggestions for Authors

It is a very interesting study focusing on evaluation of OB/GYN vulnerability. Even if the patients sample is significantly low (6 patients), the study of each case is meticulous given an important strength in this study. However, I suggest a more specific and fluent explanation in the study design. First of all you in the study design a quantitative study is described but I think it is a qualitative study. I would also suggest a better explanation of the results where  R and D results really difficult to be decoded. If you mean that is a quantitative study because each open questions (12 questions) are represented in numbers, then this is not clear from the study design. In this case I would suggest a new table.

Line 19 I would change to: “For this purpose”

Lines 35-36 I suggest you modify the phrase, maybe writing “actions or attitudes of health providers that are unfulfilled of women’s right …”

Lines 57-58 I think the message is not explained well; you could write “recognizing the psychosocial needs of women and carrying out an honest communication”

Line 61 In my opinion you should correct the verb, writing “the components of healthcare that should be improved”

Line 66 I would change the phrase in: “The main aim of this qualitative study was to evaluate the experience and perception of the women who have given birth in the last five years”. I think is a qualitative study.

Line 67 In my opinion it’s better to write “for this purpose”, instead of “for this proposed”

Line 70 If you mean from February to August, maybe it would be better to write “between February and August”.

Lines 77-78-79 I think you should explain better the concept because it’s not so clear.

Line 84 I suggest that you modify the phrase, writing “all the women were willing to participate and the informed consent…”

Line 103 I think it would be better if you modified the phrase writing “the recordings lasted”

Line 138 I suggest you add “IT” to the phrase so that it’s more clear and understandable: “they mainly associated IT with psychological impact actions” and also I think you should explain the message of the whole phrase better.

Line 275 In my opinion you should modify the order of the words in the phrase, maybe writing “The interaction systems of OB/GYN vulnerability were analysed to identify risk…”

Line 276 The phrase would sound better if you used the past form “the analysis SHOWED”

Line 329 Physical alteration?
Line 413 I suggest you explain better what you mean with “vulnerable or omissions actions”.

Line 418 “timeliness?”

Lines 438/442 I think you should rewrite these lines because the concept is not clear.

Line 445 You should use the plural form of woman, “Women”

Line 449/453 I suggest you rewrite these phrases because they’re not easily understandable.

Line 459 I think you should modify the phrase, maybe writing “adequately informing the women about breastfeeding or newborn care has shown …”

Line 465 Maybe you should write “the importance of strengthening the communication”, instead of “the importance to strengthen”

Line 474 I think you should add “that”, writing “it has been documented THAT, in pregnancy, women feel…”

Line 477 I think you should modify the phrase “And to be resilient to adapt to motherhood” because it’s not easy to understand what you meant by that.

Line 483 I think you should check this line, because in my opinion a verb is missing in the phrase.

Lines 493-494 I suggest you rewrite this phrase because it looks like there’s no principal verb, so the concept is not clear at all.

Line 502 In my opinion you could explain the phrase better, writing “is complementary to the ones already given by experts”

Line 508 I think you should write “this research analysis is not only focused on pregnancy but also postpartum…”

Line 517 You should write “the professional and the economic and technical security of the institution instead of the needs of THEMSELVES” and not “herself” because it’s referred to the women.

Line 522 I think you should use the plural form of the verb: “are key elements”

Comments on the Quality of English Language

the quality of the english language must be improved.

Author Response

It is a very interesting study focusing on evaluation of OB/GYN vulnerability. Even if the patients sample is significantly low (6 patients), the study of each case is meticulous given an important strength in this study.

Response: Thank you for the time dedicated to our manuscript and your valuable suggestions which have been incorporated into the text.

However, I suggest a more specific and fluent explanation in the study design. First of all, in the study design a quantitative study is described but I think it is a qualitative study. I would also suggest a better explanation of the results where R and D results really difficult to be decoded. If it means that it is a quantitative study because each open question (12 questions) are represented in numbers, then this is not clear from the study design. In this case I would suggest a new table.

Response: The women who participated in the in-depth interviews of the present study are part of a cohort in which psycho-emotional variables and the relationship with knowledge and perception of maternity rights were quantitatively explored (doi: 10.3390/ejihpe15020010). Women who were available for interviews and who had a moderate score (to avoid bias of experiences due to extremely traumatic or positive situations) were selected. This information has been expanded into methods section. In no case is a quantitative study by numbering the interview questions, sorry for the misunderstanding. The rooting (R) and density (D) parameters were clarified in the text (lines 120-123).

  • Line 19 I would change to: “For this purpose”
  • Lines 35-36 I suggest you modify the phrase, maybe writing “actions or attitudes of health providers that are unfulfilled of women’s right …”
  • Lines 57-58 I think the message is not explained well; you could write “recognizing the psychosocial needs of women and carrying out an honest communication”
  • Line 61 In my opinion you should correct the verb, writing “the components of healthcare that should be improved”
  • Line 66 I would change the phrase in: “The main aim of this qualitative study was to evaluate the experience and perception of the women who have given birth in the last five years”. I think it is a qualitative study.
  • Line 67 In my opinion it’s better to write “for this purpose”, instead of “for this proposed”
  • Line 70 If you mean from February to August, maybe it would be better to write “between February and August”.

Response: Thank you for your instructions, which were incorporated into the text. The study had a qualitative design, sorry for the misunderstanding.

  • Lines 77-78-79 I think you should explain better the concept because it’s not so clear.

Response: These sentences were written to increase understanding.

  • Line 84 I suggest that you modify the phrase, writing “all the women were willing to participate and the informed consent…”
  • Line 103 I think it would be better if you modified the phrase writing “the recordings lasted”
  • Line 138 I suggest you add “IT” to the phrase so that it’s more clear and understandable: “they mainly associated IT with psychological impact actions” and also I think you should explain the message of the whole phrase better.

Response: Thank you for your instructions, which were incorporated into the text. The sentence on line 138 has been improved.

  • Line 275 In my opinion you should modify the order of the words in the phrase, maybe writing “The interaction systems of OB/GYN vulnerability were analyzed to identify risk…”
  • Line 276 The phrase would sound better if you used the past form “the analysis SHOWED”
  • Line 329 Physical alteration?

Response: Thank you for your instructions, which were incorporated into the text. The concept of physical alteration was eliminated since it is not evident from the transcribed sections.

  • Line 413 I suggest you explain better what you mean with “vulnerable or omissions actions”.

Response: Thank you for this recommendation. The vulnerable and omission actions were defined (lines 413-415).

  • Line 418 “timeliness?”

Response: Timeliness of care refers to the extent to which healthcare services are provided promptly, minimizing delays that could impact patient outcomes. Timeliness is a key dimension of healthcare quality.

  • Lines 438/442 I think you should rewrite these lines because the concept is not clear.

Response: The sentences were improved to better understanding.

  • Line 445 You should use the plural form of woman, “Women”
  • Line 449/453 I suggest you rewrite these phrases because they’re not easily understandable.

Response: Thank you for your instructions, which were incorporated into the text. The sentence on line 449/453 has been improved.

  • Line 459 I think you should modify the phrase, maybe writing “adequately informing the women about breastfeeding or newborn care has shown …”
  • Line 465 Maybe you should write “the importance of strengthening the communication”, instead of “the importance to strengthen”
  • Line 474 I think you should add “that”, writing “it has been documented THAT, in pregnancy, women feel…”
  • Line 477 I think you should modify the phrase “And to be resilient to adapt to motherhood” because it’s not easy to understand what you meant by that.
  • Line 483 I think you should check this line, because in my opinion a verb is missing in the phrase.
  • Lines 493-494 I suggest you rewrite this phrase because it looks like there’s no principal verb, so the concept is not clear at all.
  • Line 502 In my opinion you could explain the phrase better, writing “is complementary to the ones already given by experts”
  • Line 508 I think you should write “this research analysis is not only focused on pregnancy but also postpartum…”
  • Line 517 You should write “the professional and the economic and technical security of the institution instead of the needs of THEMSELVES” and not “herself” because it’s referred to the women.
  • Line 522 I think you should use the plural form of the verb: “are key elements”

Response: Thank you for your recommendations, which were incorporated into the text. Furthermore, the concepts that were left without a clear understanding were reformulated, and the phrases that did not have a verb were written down.

Reviewer 2 Report

Comments and Suggestions for Authors

Dear Editor,

I am writing to express my gratitude for the opportunity to review the paper titled "Qualitative Study of Maternity Healthcare Vulnerability Based on Women's Experiences in Different Sociocultural Contexts." It was a pleasure to read and evaluate this critical work.

The authors have described their findings excellently, although the study is limited by including only six participants from Spain and Colombia. I have a few comments and suggestions that I believe could enhance the clarity and impact of the paper:

Abstract: Methods: The time and place of the study are not mentioned. Including this information would provide better context for the readers.

Introduction: I suggest providing more information on the incidence of OB/GYN vulnerability in Spain, Colombia, and other developed countries. A strong rationale for the study, highlighting gaps in previous research, would also be beneficial.

Operationalization of Concepts: The concept of "OB/GYN vulnerability" needs to be clearly defined and operationalized in the introduction section. Providing a clear framework for understanding this concept would greatly benefit the readers.

Line 66: The study is quantitative, but the abstract mentions qualitative techniques (Line 17). Please clarify or correct this discrepancy.

Sample Size and Selection: The study's sample size of six women is relatively small for concluding two countries. In the methods section, please specify that three participants are from Colombia and three from Spain and indicate whether they underwent cesarean sections or normal deliveries. Additionally, explain the selection process and address any potential biases.

Line 81: Including the mean age of the children would provide more precise information than the current age range.

Line 83: Clarify how systematic sampling was applied to select participants from a cohort of 185.

Line 90: Provide the arithmetic mean of the women's ages who participated in the study.

Line 125: The importance of triangulation analysis should be clarified. Explain the triangulation process briefly and elaborate on the theoretical framework of risk/protective factors of OB/GYN vulnerability in the methodology section.

Line 112: Specify the type of analysis performed (thematic or content analysis).

Line 113: Clarify what R=6 and D=1 indicate by using the full terms.

Line 138: Rephrase the statement for clarity, particularly the term "stable."

Lines 148-150: Differentiate whether the vulnerability statements relate to the healthcare systems in Spain or Colombia, considering the economic, social, and structural differences between the two countries.

Line 159: Make "Problems in access to healthcare" a complete statement and provide the quotation as a finding in the following line. Apply this suggestion to similar lines throughout the manuscript. Please do the same for lines 145, 162, 168, 258-268, 303, 306, 308, 311, 314, 321, 324, 332, 336, 340, 342, 349, 350, 353-355, 364-373, 384, 393-403.

Practical Implications and Feasibility: The study recommends a shift from a protocol-centered to a person-centered model in maternity healthcare. It would be helpful to clarify whether these recommendations apply to Spain and Colombia or a single country. Additionally, please provide evidence-based suggestions for minimizing burnout among health professionals and addressing biopsychosocial needs.

Line 703: Typo in mentioning the year of publication.

Thank you once again for the opportunity to review this paper. I hope my comments and suggestions help the authors refine their work.

Author Response

I am writing to express my gratitude for the opportunity to review the paper titled "Qualitative Study of Maternity Healthcare Vulnerability Based on Women's Experiences in Different Sociocultural Contexts." It was a pleasure to read and evaluate this critical work.

The authors have described their findings excellently, although the study is limited by including only six participants from Spain and Colombia. I have a few comments and suggestions that I believe could enhance the clarity and impact of the paper:

Response: Thank you for the time dedicated to our manuscript and your valuable suggestions which have been incorporated into the text.

Abstract: Methods: The time and place of the study are not mentioned. Including this information would provide better context for the readers.

Response: Time and place were incorporated into the abstract.

Introduction: I suggest providing more information on the incidence of OB/GYN vulnerability in Spain, Colombia, and other developed countries. A strong rationale for the study, highlighting gaps in previous research, would also be beneficial.

Response: The prevalence of OB/GYN vulnerability in Colombia and Spain was implemented in the introduction section (lines 41-42).

Operationalization of Concepts: The concept of "OB/GYN vulnerability" needs to be clearly defined and operationalized in the introduction section. Providing a clear framework for understanding this concept would greatly benefit the readers.

Response: The definition and operationalization of the concept have been described in the introduction section (lines 36-41).

  • Line 66: The study is quantitative, but the abstract mentions qualitative techniques (Line 17). Please clarify or correct this discrepancy.

Response: This was a qualitative study. It was already corrected in the methods section. Excuse the misunderstanding.

Sample Size and Selection: The study's sample size of six women is relatively small for concluding two countries. In the methods section, please specify that three participants are from Colombia and three from Spain and indicate whether they underwent cesarean sections or normal deliveries. Additionally, explain the selection process and address any potential biases.

Response: Despite the sample size, qualitative designs is considered the saturation of data, evidenced by the repetition of subthemes in the interviews, which is known as depth of context, something that is not achieved with a high number of interventions. The selected women who had a birth in Colombia were named as "C" while those who had a birth in Spain were named as "S". Table 1 specifies the type of birth they had (vaginal or cesarean section). It was added the availability of the women to carry out the interview and the scores in the study on perception of maternity rights (doi: 10.3390/ejihpe15020010) to select the sample.

Line 81: Including the mean age of the children would provide more precise information than the current age range.

Response: This would be fantastic; however, this study was women-focused, and the age of the offspring was not asked.

Line 83: Clarify how systematic sampling was applied to select participants from a cohort of 185.

Response: The availability of the women to carry out the interview and the scores in the study on perception of maternity rights (doi: 10.3390/ejihpe15020010) to select the sample was added.

Line 90: Provide the arithmetic mean of the women's ages who participated in the study.

Response: mean and standard deviation were added.

Line 125: The importance of triangulation analysis should be clarified. Explain the triangulation process briefly and elaborate on the theoretical framework of risk/protective factors of OB/GYN vulnerability in the methodology section.

Response: In a qualitative study, triangulation is a strategy used to enhance the credibility and reliability of research findings by using multiple sources, methods, or perspectives. Triangulation strengthens the study’s trustworthiness and provides a holistic view of the topic. Additionally, the theoretical framework of risk and protective factors of obstetric vulnerability was based on a multidisciplinary approach considering biopsychosocial model. This information was implemented in the method section.

Line 112: Specify the type of analysis performed (thematic or content analysis).

Response: This suggestion was introduced in the results section to clarify the readers.

Line 113: Clarify what R=6 and D=1 indicate by using the full terms.

Response: The clarification was added.

Line 138: Rephrase the statement for clarity, particularly the term "stable."

Response: This sentence was written to be clearer.

Lines 148-150: Differentiate whether the vulnerability statements relate to the healthcare systems in Spain or Colombia, considering the economic, social, and structural differences between the two countries.

Response: In these qualitative studies, particularly with this sample size, it would be ambitious to assume that the origin of vulnerability was focused on one socio-health context than another. With this data, we showed that experiences of vulnerability can be described in any social context, and the origin lies in health professionals and institutions.

Line 159: Make "Problems in access to healthcare" a complete statement and provide the quotation as a finding in the following line. Apply this suggestion to similar lines throughout the manuscript. Please do the same for lines 145, 162, 168, 258-268, 303, 306, 308, 311, 314, 321, 324, 332, 336, 340, 342, 349, 350, 353-355, 364-373, 384, 393-403.

Response: Thank you for this appreciation. However, they are the themes, subthemes or contexts that the accompanying transcript alludes to. It is not a phrase rather than an aid for the reader to identify the complaints, experiences and comments that the women alluded to in their interview.

Practical Implications and Feasibility: The study recommends a shift from a protocol-centered to a person-centered model in maternity healthcare. It would be helpful to clarify whether these recommendations apply to Spain and Colombia or a single country. Additionally, please provide evidence-based suggestions for minimizing burnout among health professionals and addressing biopsychosocial needs.

Response: The authors consider that the shift to a person-centered model assistance, particularly women-centered models, would be generically applicable in any socio-health context (also Colombia and Spain). In this work, we did not seek to explore the burnout of health professionals addressing biopsychosocial needs.

Line 703: Typo in mentioning the year of publication.

Response: The year for reference 51 was updated.

Round 2

Reviewer 2 Report

Comments and Suggestions for Authors

I suggest publication.

Author Response

Thank you for yout time spent reviewing our manuscript.